# Goldstone States as Non-Local Hidden Variables

Luca Fabbri 

DIME, Sez. Metodi e Modelli Matematici, Università di Genova, Via all'Opera Pia 15, 16145 Genova, Italy; fabbri@dime.unige.it

**Abstract:** We consider the theory of spinor fields in polar form, where the spinorial true degrees of freedom are isolated from their Goldstone states, and we show that these carry information about the frames which is not related to gravitation, so that their propagation is not restricted to be either causal or local: we use them to build a model of entangled spins where a singlet possesses a uniform rotation that can be made to collapse for both states simultaneously regardless their spatial distance. Models of entangled polarizations with similar properties are also sketched. An analogy with the double-slit experiment is also presented. General comments on features of Goldstone states are given.

**Keywords:** polar form; Goldstone states; non-local hidden variables

## 1. Introduction

The study of quantum mechanics, as it was developed in the first half of the last century, can claim to be a rather satisfactory theory, both in the mathematical setting and for the immense number of practical applications. Nevertheless, it is also characterized, as became clear in the second half of last century, by some problems that seem to defy full compatibility with another theory: that is, Einstein's theory of relativity.

These problems started with the work of Einstein and co-workers, culminating in the well known EPR paradox first exposed in [1]. In simple terms, the EPR paradox is what results from assuming that all the aspects of reality must be described by quantum mechanics. Because this assumption leads to a contradiction, Einstein, Podolsky, and Rosen concluded that there must be an intrinsic incompleteness in the set of observables encoded within the wave function. Therefore, the wave function contains an amount of information that is limited. What is missing is what would later be known as hidden variables.

Specifically, an EPR experiment, in the form suggested by Aharonov and Bohm [2], consists of considering particles with opposite spins and taking them apart. Because opposite spins have to conserve total angular momentum, the result of any measurement on one particle will determine the result of the measurement on the other particle, and this transfer of information is instantaneous: as the two particles have become independent due to the spatial separation, it is not possible that such a transfer of information can occur in a space-like way, and so the constant opposition of spins must be due to the fact that the two spin orientations were already chosen. Predetermination in the result of an experiment is encoded by the existence of variables present in the wave function, although hidden from the experimenter. Furthermore, these are the well-known hidden variables.

Some thirty years later, Bell proved that, if these hidden variables do indeed complement the wave function of spin singlets, then there must be a mechanism through which a measurement can influence another measurement, even if the two devices have space-like distances [3–6]. Roughly speaking, Bell showed that the existence of hidden variables completing the wave function would always imply some pattern in the results of the measurements. Such a pattern is given in the form of constraints that are known as Bell inequalities. As quantum mechanics violates these inequalities, since the results are statistically distributed, the measurements cannot have a correlation determined in the

past. Their correlation must be in the present, and as instantaneous actions for space-like separations have a non-local character; the theory must display non-locality.

It now becomes easy to assess where the problems with Einstein's relativity arise. Non-locality merely means that transmission of signals has to violate the causal structure of the light-cone. It was Bohm who suggested that such a type of faster-than-light communication does not involve any transfer of signal and consequently no incompatibility with relativity emerges [7]. An attempt to circumvent the problem by writing the de Broglie–Bohm theory in its relativistic form was by Bohm himself in reference [8].

A further source of concern for a relativistic extension comes from the subsequent enlargement to multi-particle states [9]. Since in the de Broglie–Bohm theory particles are guided by the module of the wave function, which is determined by the field equations, in multi-particle cases, the guiding equation depends on the superposed module of all particles of the universe, and the fundamental field equations are in the configuration space. The problem of the configuration space defined for multi-particle states in a relativistic setting is that in it a universal time for every particle is incompatible with relativity. Another manner to show this fact is to observe that the guidance equation is given in terms of the velocity, whereas the theory gives only the velocity operator so that with the wave function we can only compute the velocity density. The velocity in itself can be computed either by integrating the velocity density over the volume, as in the Ehrenfest theorem, or by dividing by the density distribution, that is the norm of the wave function. Volume integrals might be extended to curvilinear coordinates only for scalar quantities. Then, the density distribution in the relativistic case is $\overline{\psi}\psi$ and is not positive, while a positive density distribution would be given by $\psi^{\dagger}\psi$, although this is not a scalar [10]. Hence, neither volume integration nor density quotients preserve manifest covariance, and the problems with the relativistic extensions of the de Broglie–Bohm formulation persist.

One solution may be abandoning configuration space, with progress for the study of spin singlets [11]. However, non-local actions still ask for non-relativistic treatments.

So, another way out may then consist of abandoning the idea of particles, however many they are. We will not aim to discuss whether particles can be treated as localized wave functions. However, we shall consider the wave function as the only object encoding the description of a quantum mechanical process. To avoid issues of generalizations, we will consider immediately the relativistic case. Furthermore, to be in the most complete situation, we will also consider spin from the start. That is, we will consider the spinor field theory in its most general case [12]. However we are going to take it in polar form [13,14]. In this way, the spinorial true degrees of freedom are isolated from the components that can be seen as the spinorial equivalent of the well-known Goldstone states [15–18].

We will demonstrate that these Goldstone states can encode some information as non-local hidden variables for pairs of entangled spins.

Analogies, for photons, with entangled polarizations, are discussed.

A parallel with the two-slit experiment is given.

## 2. Physical Fields in Polar Form

We begin by considering the spinor field theory in polar decomposition: we will not present it as it was originally presented in [13,14], but in a manifestly covariant manner [15]. In parallel, we will also consider the vector field in polar decomposition [16]. The fact that vectors are real might suggest that there can not be a polar form in analogy to the one of spinors since these are complex, but we will see that such a decomposition is possible for both.

*2.1. Kinematic Quantities*

2.1.1. Transformation Laws and Fundamental Fields

We begin by assuming the existence of a pair of inverse metric tensors $g_{\mu\nu} = g_{\nu\mu}$ and $g^{\mu\nu} = g^{\nu\mu}$ with $g_{\mu\rho}g^{\rho\alpha} = \delta^\alpha_\mu$, where $\delta^\alpha_\mu$ is the Kronecker delta. The pair of dual tetrads

$$\xi^\mu_a \xi^b_\mu = \delta^b_a \qquad\qquad \xi^\mu_a \xi^a_\nu = \delta^\mu_\nu \tag{1}$$

verify the ortho-normalization conditions

$$\xi^a_\mu \xi^b_\nu g^{\mu\nu} = \eta^{ab} \qquad \xi^\mu_a \xi^\nu_b g_{\mu\nu} = \eta_{ab} \tag{2}$$

with $\eta = \eta^T = \eta^{-1}$ the Minkowskian matrix. We introduce also the Clifford matrices $\gamma^a$ verifying the relations

$$\{\gamma^a, \gamma^b\} = 2\mathbb{I}\eta^{ab} \tag{3}$$

where $\mathbb{I}$ is the identity matrix and

$$\tfrac{1}{4}[\gamma^a, \gamma^b] = \sigma^{ab} \tag{4}$$

such that

$$2i\sigma_{ab} = \varepsilon_{abcd}\boldsymbol{\pi}\sigma^{cd} \tag{5}$$

being $\varepsilon_{abcd}$ the antisymmetric pseudo-tensor and implicitly defining the $\boldsymbol{\pi}$ matrix (this is usually denoted as a gamma with index five, but it has no sense in the space-time and so we use a notation with no index). The Greek indices are associated with general coordinate transformations, or passive transformations. Instead, the Latin indices are associated with Lorentz transformations, or active transformations. In fact, these are the transformations shuffling vectors within the basis of tetrads and so this transformation must be a Lorentz transformation since we need preserve the Minkowskian matrix. We have that such Lorentz transformations can be written as follows:

$$(\Lambda)^a{}_b = \exp\left(\theta^a{}_b\right) \tag{6}$$

where we have that $\theta_{ab} = -\theta_{ba}$ are the parameters of the transformation. In this form they are in real representation. Nevertheless, they might also be written in complex representation. The complex representation of a Lorentz transformation is given according to

$$\boldsymbol{\Lambda} = \exp\left(\tfrac{1}{2}\sigma^{ab}\theta_{ab}\right) \tag{7}$$

with $\sigma^{ab}$ given by (4) but $\theta_{ab} = -\theta_{ba}$ are the parameters of the transformation exactly as above. Of both real and complex representations, we can provide an explicit form. The real Lorentz transformations can be written as

$$(\Lambda)^a{}_b = \left(X^2 + Y^2 + \tfrac{1}{8}Z^{ij}Z_{ij}\right)\delta^a_b - \tfrac{1}{2}Z^{ak}Z_{bk} + \left(\tfrac{1}{2}YZ_{ij}\varepsilon^{ijak} - XZ^{ak}\right)\eta_{kb} \tag{8}$$

in terms of $X$, $Y$ and $Z_{ab}$ functions. The complex Lorentz transformations can be written as

$$\boldsymbol{\Lambda} = X\mathbb{I} + Yi\boldsymbol{\pi} + \tfrac{1}{2}Z^{ab}\sigma_{ab} \tag{9}$$

in terms of the same $X$, $Y$ and $Z_{ab}$ functions. These functions are defined by

$$X := \cos y \cosh x \tag{10}$$

$$Y := \sin y \sinh x \tag{11}$$

$$Z^{ab} := \left( \frac{x \sinh x \cos y + y \sin y \cosh x}{x^2 + y^2} \right) \theta^{ab} + \left( \frac{x \cosh x \sin y - y \cos y \sinh x}{x^2 + y^2} \right) \frac{1}{2} \theta_{ij} \varepsilon^{ijab} \tag{12}$$

such that $8X^2 - 8Y^2 + Z^{ab}Z_{ab} \equiv 8$ and $32XY - Z^{ij}Z^{ab}\varepsilon_{ijab} \equiv 0$ whose arguments are such that

$$2x^2 := a + \sqrt{a^2 + b^2} \tag{13}$$

$$2y^2 := -a + \sqrt{a^2 + b^2} \tag{14}$$

where

$$a := -\tfrac{1}{8}\theta_{ij}\theta^{ij} \tag{15}$$

$$b := \tfrac{1}{16}\theta_{ij}\theta_{ab}\varepsilon^{ijab} \tag{16}$$

and with $\theta_{ij}$ being the parameters of the Lorentz transformations. Therefore, the knowledge of the parameters $\theta_{ij}$ allows to compute through (15) and (16) the $a$ and $b$ squared-parameters. These are used to calculate via (13) and (14) the $x$ and $y$ arguments of the $X$, $Y$, and $Z_{ab}$ functions. Furthermore, with these functions (8) and (9) give the explicit form of the real and complex Lorentz transformations. To see that these are indeed Lorentz transformations, we can check specific examples. For a single boost with $\theta_{03} = -\varphi$, we have that $y = 0$ and $x = |\varphi/2|$, so that

$$\cosh(\varphi/2) = X \tag{17}$$

$$-2\sinh(\varphi/2) = Z_{03} \tag{18}$$

from which

$$(\Lambda_{B3})^a{}_b = \delta^a_b - (\delta^a_3\delta^3_b + \delta^a_0\delta^0_b)(1 - \cosh\varphi) + (\delta^a_3\delta^0_b + \delta^a_0\delta^3_b)\sinh\varphi \tag{19}$$

or component-by-component

$$\Lambda_{B3} = \left( \begin{array}{c|ccc} \cosh\varphi & 0 & 0 & \sinh\varphi \\ \hline 0 & 1 & 0 & 0 \\ 0 & 0 & 1 & 0 \\ \sinh\varphi & 0 & 0 & \cosh\varphi \end{array} \right) \tag{20}$$

as is known. The same occurs for every boost. For a single rotation $\theta_{12} = -\theta$, so that $x = 0$ and $y = |\theta/2|$, and finally

$$\cos(\theta/2) = X \tag{21}$$

$$-2\sin(\theta/2) = Z_{12} \tag{22}$$

yielding

$$(\Lambda_{R3})^a{}_b = \delta^a_b - (\delta^a_1\delta^1_b + \delta^a_2\delta^2_b)(1 - \cos\theta) + (\delta^a_2\delta^1_b - \delta^a_1\delta^2_b)\sin\theta \tag{23}$$

or component-by-component

$$\Lambda_{R3} = \left( \begin{array}{c|ccc} 1 & 0 & 0 & 0 \\ \hline 0 & \cos\theta & \sin\theta & 0 \\ 0 & -\sin\theta & \cos\theta & 0 \\ 0 & 0 & 0 & 1 \end{array} \right) \tag{24}$$

as is also known. Furthermore, the same for each rotation. In the case of the complex Lorentz transformations, a single boost is, for instance, as follows:

$$\mathbf{\Lambda}_{B3} = \begin{pmatrix} e^{\varphi/2} & 0 & 0 & 0 \\ 0 & e^{-\varphi/2} & 0 & 0 \\ 0 & 0 & e^{-\varphi/2} & 0 \\ 0 & 0 & 0 & e^{\varphi/2} \end{pmatrix} \tag{25}$$

with the same parameter as above. A single rotation is, for instance, as follows:

$$\mathbf{\Lambda}_{R3} = \begin{pmatrix} e^{i\theta/2} & 0 & 0 & 0 \\ 0 & e^{-i\theta/2} & 0 & 0 \\ 0 & 0 & e^{i\theta/2} & 0 \\ 0 & 0 & 0 & e^{-i\theta/2} \end{pmatrix} \tag{26}$$

again with the same parameter as above. Of course the proof that (8) and (9) are real and complex Lorentz transformations can also be carried out in full generality. To see that, consider (8) and take its inverse

$$(\Lambda^{-1})^i{}_j = \left(X^2 + Y^2 + \tfrac{1}{8}Z^{cd}Z_{cd}\right)\delta^i_j - \tfrac{1}{2}Z_{jc}Z^{ic} - \left(\tfrac{1}{2}YZ_{cd}\varepsilon^{cdiq} - XZ^{iq}\right)\eta_{qj}$$

$$= \eta^{ib}[\left(X^2 + Y^2 + \tfrac{1}{8}Z^{cd}Z_{cd}\right)\delta^a_b - \tfrac{1}{2}Z^{ac}Z_{bc} + \left(\tfrac{1}{2}YZ_{cd}\varepsilon^{cdap} - XZ^{ap}\right)\eta_{pb}]\eta_{aj} = \eta^{ib}(\Lambda^T)_b{}^a\eta_{aj} \tag{27}$$

which then gives

$$(\Lambda)^a{}_k(\Lambda)^b{}_i\eta^{ki} = \eta^{ab} \tag{28}$$

showing that it preserves the Minkowskian matrix and so it is a Lorentz transformation in real representation. In the other case, take into account (9) whose inverse is given as

$$\mathbf{\Lambda}^{-1} = X\mathbb{I} + Yi\boldsymbol{\pi} - \tfrac{1}{2}Z^{ab}\sigma_{ab} \tag{29}$$

from which

$$\mathbf{\Lambda}\gamma^b\mathbf{\Lambda}^{-1} = (X\mathbb{I} + Yi\boldsymbol{\pi} + \tfrac{1}{2}Z^{pq}\sigma_{pq})\gamma^b(X\mathbb{I} + Yi\boldsymbol{\pi} - \tfrac{1}{2}Z^{ij}\sigma_{ij}) = [(X^2 + Y^2 + \tfrac{1}{8}Z^{ij}Z_{ij})\delta^b_a$$

$$- \tfrac{1}{2}Z_{ak}Z^{bk} + (\tfrac{1}{2}YZ^{ij}\varepsilon_{ijka} - XZ_{ka})\eta^{kb}]\gamma^a + \tfrac{i}{4}(\tfrac{1}{4}Z_{ij}Z_{pq}\varepsilon^{ijpq}\delta^b_a + Z^{bk}Z^{ij}\varepsilon_{ijka})\gamma^a\boldsymbol{\pi}$$

$$= [(X^2 + Y^2 + \tfrac{1}{8}Z^{ij}Z_{ij})\delta^b_a - \tfrac{1}{2}Z_{ak}Z^{bk} - (\tfrac{1}{2}YZ^{ij}\varepsilon_{ijak} - XZ_{ak})\eta^{kb}]\gamma^a = (\Lambda^{-1})^b_a\gamma^a \tag{30}$$

and finally

$$(\Lambda)^a{}_b\mathbf{\Lambda}\gamma^b\mathbf{\Lambda}^{-1} = \gamma^a \tag{31}$$

showing that this is the Lorentz transformation in complex representation. Therefore, (8) and (9) are the explicit form of the real and complex representations of the Lorentz transformations. To conclude, we highlight that, once a complex Lorentz transformation is given, it is possible to combine it with a generic phase, as

$$\mathbf{S} = \mathbf{\Lambda}e^{iq\alpha} = (X\mathbb{I} + Yi\boldsymbol{\pi} + \tfrac{1}{2}Z^{ab}\sigma_{ab})e^{iq\alpha} \tag{32}$$

which is the most complete transformation, known as spinorial transformation.

The transformations (8) and (9) are the basis upon which to build the fundamental fields, since in physics the fundamental fields are defined as what transforms in terms of a given transformation law. So, any column of 4 real functions transforming according to

$$V^a \to (\Lambda)^a{}_b V^b \tag{33}$$

is called vector field. Similarly, any column of 4 complex functions transforming as

$$\psi \to S\psi \tag{34}$$

is called the spinor field. Definitions can be given in more general cases, but this is all we need.

It is possible to vertically move indices by means of

$$V_a = \eta_{ab} V^b \qquad V^i = \eta^{ij} V_j \tag{35}$$

which is the transposition of a vector. This procedure is essential to set $V^2 = V_a V^a$ as scalar product. In addition, see

$$\overline{\psi} = \psi^\dagger \gamma^0 \qquad \psi = \gamma^0 \overline{\psi}^\dagger \tag{36}$$

as the adjunction of a spinor. With such a pair of adjoint spinors we define the following bi-linear spinor quantities

$$\Sigma^{ab} := 2\overline{\psi}\sigma^{ab}\pi\psi \tag{37}$$

$$M^{ab} := 2i\overline{\psi}\sigma^{ab}\psi \tag{38}$$

$$S^a := \overline{\psi}\gamma^a\pi\psi \tag{39}$$

$$U^a := \overline{\psi}\gamma^a\psi \tag{40}$$

$$\Theta := i\overline{\psi}\pi\psi \tag{41}$$

$$\Phi := \overline{\psi}\psi \tag{42}$$

which are all real tensors, such that

$$\psi\overline{\psi} \equiv \tfrac{1}{4}\Phi\mathbb{I} + \tfrac{1}{4}U_a\gamma^a + \tfrac{i}{8}M_{ab}\sigma^{ab} - \tfrac{1}{8}\Sigma_{ab}\sigma^{ab}\pi - \tfrac{1}{4}S_a\gamma^a\pi - \tfrac{i}{4}\Theta\pi \tag{43}$$

as well as

$$\Sigma^{ab} \equiv -\tfrac{1}{2}\varepsilon^{abij}M_{ij} \tag{44}$$

with

$$M_{ab}\Theta + \Sigma_{ab}\Phi \equiv U_{[a}S_{b]} \tag{45}$$

alongside to

$$U_a S^a \equiv 0 \tag{46}$$

$$U_a U^a \equiv -S_a S^a \equiv \Theta^2 + \Phi^2 \tag{47}$$

as is straightforward to prove, called Fierz identities.

2.1.2. Polar Decompositions

Because the fundamental fields are defined in terms of their transformation laws, it is possible to employ these transformations to write the fields in ways that are somewhat special. To see how, let us start by considering the vector field in its most general form. It is possible to see that one can always write the vector according to

$$V^a = \phi v^a \tag{48}$$

where $\phi$ is a real scalar field and the only degree of freedom, known as module. Notice that we can have all cases given by $v^2 = 1$, $v^2 = 0$ or $v^2 = -1$ in general. For a spinor field in its

most general form it is possible to demonstrate a similar result. When $\Theta^2 + \Phi^2 \neq 0$, we have that one can always write the spinor according to the form

$$\psi = \phi e^{-\frac{i}{2}\beta\pi} L^{-1} \begin{pmatrix} 1 \\ 0 \\ 1 \\ 0 \end{pmatrix} \tag{49}$$

in chiral representation, with $L$ a Lorentz transformation and with $\phi$ and $\beta$ real scalar and pseudo-scalar fields and the only degrees of freedom, known as module and chiral angle. In this form the bi-linear spinor quantities are expressed explicitly as

$$\Sigma^{ab} = 2\phi^2 (\cos\beta\, u^{[a} s^{b]} - \sin\beta\, u_j s_k \varepsilon^{jkab}) \tag{50}$$

$$M^{ab} = 2\phi^2 (\cos\beta\, u_j s_k \varepsilon^{jkab} + \sin\beta\, u^{[a} s^{b]}) \tag{51}$$

with

$$S^a = 2\phi^2 s^a \tag{52}$$

$$U^a = 2\phi^2 u^a \tag{53}$$

and

$$\Theta = 2\phi^2 \sin\beta \tag{54}$$

$$\Phi = 2\phi^2 \cos\beta \tag{55}$$

from which

$$\psi\overline{\psi} \equiv \tfrac{1}{2}\phi^2 e^{-i\beta\pi} (e^{i\beta\pi} + u_a\gamma^a)(e^{-i\beta\pi} - s_a\gamma^a\pi) \tag{56}$$

and

$$u_a s^a \equiv 0 \tag{57}$$

$$u_a u^a \equiv -s_a s^a \equiv 1 \tag{58}$$

are the normalized velocity vector and spin axial-vector, as is well known. Written in polar form, the 8 real components of the spinor can be rearranged in such a way that the 2 real scalar degrees of freedom are isolated from the 6 real components that can always be transferred into the frame through the 6 parameters of the Lorentz transformation. Thus, $L$ encodes what is defined to be the spinorial equivalent of Goldstone states (when the special situation $\Theta^2 + \Phi^2 \equiv 0$ occurs an analogous polar decomposition can be done, although this case is constituted by the singular spinor fields. Because this might mean that these specific fields are non-physical, we are not going to consider them in the following of the present work). Readers interested in a more detailed discussion might have a look at [16] and specifically for spinors at [15].

### 2.1.3. Covariant Derivatives

In what we have shown so far, we have considered all transformations as local and fields as point-dependent. Hence, we should expect some gauge connection to appear in the covariant derivatives. To see how, let us then set the form of the covariant derivative as

$$\nabla_\mu V^a = \partial_\mu V^a + \Omega^a{}_{b\mu} V^b \tag{59}$$

in terms of the spin connection. Analogously, we have

$$\boldsymbol{\nabla}_\mu \psi = \partial_\mu \psi + \tfrac{1}{2}\Omega_{ij\mu}\sigma^{ij}\psi + iqA_\mu\psi \tag{60}$$

in terms of spin connection and gauge potential. General definitions can be taken from fundamental textbooks, for instance, [12].

### 2.1.4. Tensorial Connections

We now consider again the transformation in their explicit form (8) and (9). With a straightforward calculation and considering the identities given by $8X^2 - 8Y^2 + Z^{ab}Z_{ab} \equiv 8$ and $32XY - Z^{ij}Z^{ab}\varepsilon_{ijab} \equiv 0$ one can see that it is always possible to write

$$(\Lambda)^i{}_k \partial_\mu (\Lambda^{-1})^k{}_j := \partial_\mu \xi^i{}_j \tag{61}$$

for some $\xi^i{}_j$ which is defined to be the Goldstone state. Similarly,

$$\boldsymbol{L}^{-1} \partial_\mu \boldsymbol{L} := iq \partial_\mu \xi \mathbb{I} + \frac{1}{2} \partial_\mu \xi^{ab} \sigma_{ab} \tag{62}$$

for some $\xi$ and $\xi^{ab}$ defined to be the Goldstone states.

When the covariant derivatives are taken for the fields in polar form, and these definitions are used, it is easy to see that, upon introduction of

$$\partial_\mu \xi^i{}_j - \Omega^i{}_{j\mu} := R^i{}_{j\mu} \tag{63}$$

we can write

$$\nabla_\mu V^a = (\delta^a_b \nabla_\mu \ln \phi - R^a{}_{b\mu}) V^b \tag{64}$$

for the vector field. Analogously, defining

$$\partial_\mu \xi_{ij} - \Omega_{ij\mu} := R_{ij\mu} \tag{65}$$
$$q(\partial_\mu \xi - A_\mu) := P_\mu \tag{66}$$

we can write

$$\boldsymbol{\nabla}_\mu \psi = \left(-\frac{i}{2} \nabla_\mu \beta \boldsymbol{\pi} + \nabla_\mu \ln \phi \mathbb{I} - i P_\mu \mathbb{I} - \frac{1}{2} R_{ij\mu} \sigma^{ij}\right) \psi \tag{67}$$

for the spinor field. Notice that the Goldstone states are absorbed by spin connection and gauge potential as the longitudinal components of $P_\mu$ and $R_{ji\mu}$ which are a real vector and tensor, respectively. As such, they encode the same information of gauge potential and spin connection but they are gauge invariant and covariant. In the rest of the presentation, we will call these tensorial connections.

To conclude this sub-section, we notice that

$$\nabla_\mu v_a = R_{ba\mu} v^b \tag{68}$$

holds as a general identity. Furthermore, analogously

$$\nabla_\mu s_i = R_{ji\mu} s^j \tag{69}$$
$$\nabla_\mu u_i = R_{ji\mu} u^j \tag{70}$$

is also a general identity. The similarities of these identities are profound.

Readers interested in details may have a look at [15,16].

### 2.1.5. Curvatures

As a final remark, we can see, via some straightforward computation of the commutators of covariant derivatives, that the Riemann curvature and Maxwell strength are

$$R^i_{\ j\mu\nu} \equiv -(\nabla_\mu R^i_{\ j\nu} - \nabla_\nu R^i_{\ j\mu} + R^i_{\ k\mu}R^k_{\ j\nu} - R^i_{\ k\nu}R^k_{\ j\mu}) \tag{71}$$

$$qF_{\mu\nu} \equiv -(\nabla_\mu P_\nu - \nabla_\nu P_\mu) \tag{72}$$

identically. These expressions are important because they show that there is a direct link between tensorial connections and their curvatures. Specifically, one can consider the problem of asking whether the conditions $R^i_{\ j\mu\nu} = 0$ and $F_{\mu\nu} = 0$ have non-zero solution, and the answer is yes [15]. Hence, it is possible to have situations in which physical effects are non-trivial, albeit determined by sourceless fields. The situation thus described might look anomalous, but actually there already are examples of physical circumstances in which this naturally happens. The Aharonov–Bohm effect is precisely one of them.

### 2.2. Dynamical Coupling
### 2.2.1. Field Equations

Having introduced the kinematic quantities, it is now time to have them coupled. The dynamics of the vector is assigned in terms of the Proca equations

$$\nabla_\sigma (\partial V)^{\sigma\mu} + M^2 V^\mu = \Gamma^\mu \tag{73}$$

with $(\partial V)_{\sigma\mu} = \partial_\sigma V_\mu - \partial_\mu V_\sigma$ and $\Gamma^\mu$ being a generic external source. The dynamics of the spinor is determined by the Dirac equations given by

$$i\gamma^\mu \boldsymbol{\nabla}_\mu \psi - X W_\mu \gamma^\mu \boldsymbol{\pi} \psi - m\psi = 0 \tag{74}$$

with $W_\mu$ axial-vector torsion and $X$ torsion-spin coupling added to be in the most general situation possible [12].

### 2.2.2. Polar Form

Writing the above equations in polar form is now immediately done. For the Proca equations we have

$$(g^{\alpha\nu}\nabla^2\phi - \nabla^\nu\nabla^\alpha\phi - R^{\nu\mu}_{\ \ \mu}\nabla^\alpha\phi + R^{\nu\alpha\sigma}\nabla_\sigma\phi + R^{\nu[\alpha\sigma]}\nabla_\sigma\phi +$$
$$+ \nabla_\sigma R^{\nu[\alpha\sigma]}\phi + R^{\sigma[\alpha\pi]}R^\nu_{\ \sigma\pi}\phi + M^2 g^{\alpha\nu}\phi)v_\nu = \Gamma^\alpha \tag{75}$$

as was shown in [16]. For the Dirac equations

$$B_\mu - 2P^\iota u_{[\iota} s_{\mu]} + (\nabla\beta - 2XW)_\mu + 2s_\mu m \cos\beta = 0 \tag{76}$$

$$R_\mu - 2P^\rho u^\nu s^\alpha \varepsilon_{\mu\rho\nu\alpha} + 2s_\mu m \sin\beta + \nabla_\mu \ln\phi^2 = 0 \tag{77}$$

with $R_{\mu a}^{\ \ a} = R_\mu$ and $\frac{1}{2}\varepsilon_{\mu a\nu\iota}R^{a\nu\iota} = B_\mu$ and because these are merely two Gordon decompositions together implying the Dirac equations, then they are equivalent to the Dirac equations themselves. These are 8 real equations exactly as the pair of vector Equations (76) and (77) above.

Readers interested in details can find them in [15,16].

## 3. Goldstone States
### 3.1. Entangled Observables

Having converted the theories in polar form, it is time to see the advantages of such a formalism by looking for specific solutions. Furthermore, particularly interesting for us

will be the solutions with the structure shown in [15]. Hence the tensorial connection is selected to be

$$R_{r\theta\theta} = -r \tag{78}$$

$$R_{r\varphi\varphi} = -r|\sin\theta|^2 \tag{79}$$

$$R_{\theta\varphi\varphi} = -r^2\cos\theta\sin\theta \tag{80}$$

$$R_{rtt} = -2\varepsilon\sinh\alpha \tag{81}$$

$$R_{\varphi rt} = 2\varepsilon r\sin\theta\cosh\alpha \tag{82}$$

$$P_t = m \tag{83}$$

with $\varepsilon$ constant, such that $m > \varepsilon > 0$ giving zero Riemann curvature and zero Maxwell strength. A spin axial-vector and a velocity vector compatible with the above are

$$s_\theta = -r \tag{84}$$

$$u_t = \cosh\alpha \tag{85}$$

$$u_\varphi = r\sin\theta\sinh\alpha \tag{86}$$

with $\sinh\alpha = \sqrt{\varepsilon(2m-\varepsilon)}/(m-\varepsilon)$ as necessary constraint on the $\alpha$ constant. In fact, in this way, we can see that

$$\beta = 0 \tag{87}$$

$$\phi^2 r^2 \sin\theta = K^2 e^{-2r\sqrt{\varepsilon(2m-\varepsilon)}} \tag{88}$$

with $K$ a generic integration constant. All these elements concur to have the polar Equations (76) and (77) verified, as it can easily be checked with a straightforward substitution.

We will now have the above solution written according to the usual spinorial form. However, it is possible to see that the same solution in polar form generates a two-fold multiplicity of solutions in the usual spinor form. In fact, we have that one solution can be written according to

$$\psi = \frac{K}{r\sqrt{\sin\theta}} e^{-r\sqrt{\varepsilon(2m-\varepsilon)}} e^{-it(m-\varepsilon)} \begin{pmatrix} 1 \\ 0 \\ 1 \\ 0 \end{pmatrix} \tag{89}$$

with tetrads

$$\xi_0^t = \cosh\alpha \quad \xi_2^t = -\sinh\alpha \tag{90}$$

$$\xi_1^r = -1 \tag{91}$$

$$\xi_3^\theta = \frac{1}{r} \tag{92}$$

$$\xi_0^\varphi = -\frac{1}{r\sin\theta}\sinh\alpha \quad \xi_2^\varphi = \frac{1}{r\sin\theta}\cosh\alpha \tag{93}$$

giving spin connection

$$\Omega_{13\theta} = -1 \tag{94}$$

$$\Omega_{01\varphi} = -\sin\theta\sinh\alpha \tag{95}$$

$$\Omega_{03\varphi} = \cos\theta\sinh\alpha \tag{96}$$

$$\Omega_{12\varphi} = -\sin\theta\cosh\alpha \tag{97}$$

$$\Omega_{23\varphi} = -\cos\theta\cosh\alpha \tag{98}$$

with zero Riemann curvature. All these elements concur to have the Dirac Equation (74) verified. As is clear, such solution corresponds to the spin-up case. Nevertheless, an alternative solution can be written according to

$$\psi = \frac{K}{r\sqrt{\sin\theta}} e^{-r\sqrt{\varepsilon(2m-\varepsilon)}} e^{-it(m-\varepsilon)} \begin{pmatrix} 0 \\ 1 \\ 0 \\ 1 \end{pmatrix} \tag{99}$$

with tetrads

$$\xi_0^t = \cosh\alpha \quad \xi_2^t = -\sinh\alpha \tag{100}$$

$$\xi_1^r = 1 \tag{101}$$

$$\xi_3^\theta = -\frac{1}{r} \tag{102}$$

$$\xi_0^\varphi = -\frac{1}{r\sin\theta}\sinh\alpha \quad \xi_2^\varphi = \frac{1}{r\sin\theta}\cosh\alpha \tag{103}$$

giving spin connection

$$\Omega_{13\theta} = -1 \tag{104}$$

$$\Omega_{01\varphi} = \sin\theta\sinh\alpha \tag{105}$$

$$\Omega_{03\varphi} = -\cos\theta\sinh\alpha \tag{106}$$

$$\Omega_{12\varphi} = \sin\theta\cosh\alpha \tag{107}$$

$$\Omega_{23\varphi} = \cos\theta\cosh\alpha \tag{108}$$

with zero Riemann curvature. All these elements concur to have the Dirac Equation (74) verified. Furthermore, as is clear, this solution corresponds to the spin-down case. The fact that we have generated a pair of solutions with opposite spin orientation is the consequence of the double-helicity structure of the spinor field encoded in its very definition.

Nevertheless, this is not the full amount of information we can extract. Additionally, in fact, one can also have

$$\psi = \frac{K}{r\sqrt{\sin\theta}} e^{-r\sqrt{\varepsilon(2m-\varepsilon)}} e^{-it(m-\varepsilon)} \begin{pmatrix} \cos\zeta/2 \\ -\sin\zeta/2 \\ \cos\zeta/2 \\ -\sin\zeta/2 \end{pmatrix} \tag{109}$$

with tetrads

$$\xi_0^t = \cosh\alpha \quad \xi_2^t = -\sinh\alpha \tag{110}$$

$$\xi_1^r = -\cos\zeta \quad \xi_3^r = -\sin\zeta \tag{111}$$

$$\xi_1^\theta = -\frac{1}{r}\sin\zeta \quad \xi_3^\theta = \frac{1}{r}\cos\zeta \tag{112}$$

$$\xi_0^\varphi = -\frac{1}{r\sin\theta}\sinh\alpha \quad \xi_2^\varphi = \frac{1}{r\sin\theta}\cosh\alpha \tag{113}$$

and spin connection

$$\Omega_{13t} = -\omega \tag{114}$$

$$\Omega_{13\theta} = -1 \tag{115}$$

$$\Omega_{01\varphi} = -\sin(\theta+\zeta)\sinh\alpha \tag{116}$$

$$\Omega_{03\varphi} = \cos(\theta+\zeta)\sinh\alpha \tag{117}$$

$$\Omega_{12\varphi} = -\sin(\theta+\zeta)\cosh\alpha \tag{118}$$

$$\Omega_{23\varphi} = -\cos(\theta+\zeta)\cosh\alpha \tag{119}$$

with $\zeta = \omega t$ and $\omega$ constant and verifying the Dirac equations identically. Furthermore, analogously we also have that

$$\psi = \frac{K}{r\sqrt{\sin\theta}} e^{-r\sqrt{\varepsilon(2m-\varepsilon)}} e^{-it(m-\varepsilon)} \begin{pmatrix} \sin\zeta/2 \\ \cos\zeta/2 \\ \sin\zeta/2 \\ \cos\zeta/2 \end{pmatrix} \tag{120}$$

with tetrads

$$\xi_0^t = \cosh\alpha \quad \xi_2^t = -\sinh\alpha \tag{121}$$

$$\xi_1^r = \cos\zeta \quad \xi_3^r = \sin\zeta \tag{122}$$

$$\xi_1^\theta = \frac{1}{r}\sin\zeta \quad \xi_3^\theta = -\frac{1}{r}\cos\zeta \tag{123}$$

$$\xi_0^\varphi = -\frac{1}{r\sin\theta}\sinh\alpha \quad \xi_2^\varphi = \frac{1}{r\sin\theta}\cosh\alpha \tag{124}$$

and spin connection

$$\Omega_{13t} = -\omega \tag{125}$$

$$\Omega_{13\theta} = -1 \tag{126}$$

$$\Omega_{01\varphi} = \sin(\theta+\zeta)\sinh\alpha \tag{127}$$

$$\Omega_{03\varphi} = -\cos(\theta+\zeta)\sinh\alpha \tag{128}$$

$$\Omega_{12\varphi} = \sin(\theta+\zeta)\cosh\alpha \tag{129}$$

$$\Omega_{23\varphi} = \cos(\theta+\zeta)\cosh\alpha \tag{130}$$

always, with $\zeta = \omega t$ and $\omega$ constant, and verifying the Dirac equations identically. Hence, while still maintaining their spin opposition, the two spinors have their spin axis also displaying a rotation with angle $\zeta = \omega t$ and $\omega$ constant.

Nevertheless, any observation that breaks the rotation fixing $\omega = 0$ has an effect. In fact, suppose the observation is performed at a time such that $\omega t \approx 2n\pi$ then the solution (109)–(119) are given by (89)–(98) plus the $\Omega_{13t} = -\omega$ condition, and (120)–(130) are (99)–(108) plus the $\Omega_{13t} = -\omega$ condition. Now, if the system were disturbed so that $\omega = 0$, then the rotation would stop, simultaneously locking the first solution to the spin-up state and the second solution to the spin-down state. If instead it is $\omega t \approx (2n+1)\pi$, then the solution (109)–(119) would be (99)–(108) plus the $\Omega_{13t} = -\omega$ condition, and (120)–(130) are (89)–(98) plus the $\Omega_{13t} = -\omega$ condition. Now, if the system were disturbed so that $\omega = 0$, then the rotation would stop, simultaneously locking the first solution to spin-down states and the second solution to spin-up states. Either way, any observation that stops the rotation has the effect of making both spinors locked to either one of the two definite spin states concurrently.

The simultaneous collapse of both spinors does not depend on this particular type of solution. In fact, the two solutions (109)–(119) and (120)–(130) are what results after applying to the two solutions (89)–(98) and (99)–(108) the rotation given in its complex representation according to

$$R^{-1} = \begin{pmatrix} \cos\zeta/2 & \sin\zeta/2 & 0 & 0 \\ -\sin\zeta/2 & \cos\zeta/2 & 0 & 0 \\ 0 & 0 & \cos\zeta/2 & \sin\zeta/2 \\ 0 & 0 & -\sin\zeta/2 & \cos\zeta/2 \end{pmatrix} \tag{131}$$

and in its real representation according to

$$(R^{-1})^a{}_b = \begin{pmatrix} 1 & 0 & 0 & 0 \\ 0 & \cos\zeta & 0 & -\sin\zeta \\ 0 & 0 & 1 & 0 \\ 0 & \sin\zeta & 0 & \cos\zeta \end{pmatrix} \tag{132}$$

where $\zeta = \omega t$ and $\omega$ is a generic constant. This allows for a generalization of the previous analysis. In fact, one could see that already for the spinors in their most general form (49) it is possible to specify only the rotation as

$$\psi = \phi e^{-\frac{i}{2}\beta\pi} e^{-iq\alpha} \begin{pmatrix} \cos\zeta/2 \\ -\sin\zeta/2 \\ \cos\zeta/2 \\ -\sin\zeta/2 \end{pmatrix} \tag{133}$$

giving $s^3 = \cos\zeta$ as well as

$$\psi = \phi e^{-\frac{i}{2}\beta\pi} e^{-iq\alpha} \begin{pmatrix} \sin\zeta/2 \\ \cos\zeta/2 \\ \sin\zeta/2 \\ \cos\zeta/2 \end{pmatrix} \tag{134}$$

giving $s^3 = -\cos\zeta$ and so that the two spin states are in constant opposition, and with uniform flipping over time.

As is clear from this analysis, the specific spinor is not determinant, and what is important is only the structure of the complex rotation $\boldsymbol{R}^{-1}$ in general. Because

$$\boldsymbol{R}^{-1}\partial_t\boldsymbol{R} = -\omega\sigma_{13} \tag{135}$$

we have that (62) yields

$$\partial_t\xi_{13} = -\omega \tag{136}$$

as the Goldstone states. Since $\Omega_{13t} = \partial_t\xi_{13}$, we obtain

$$\Omega_{13t} = -\omega \tag{137}$$

as the additional component of the spin connection, as it was found above, showing that the additional component (114)–(125) of the spin connection does not depend on the special solution, it is a general character of (131). Notice that the Goldstone states are not observable states as is expected in any gauge covariant theory, and as is known from the standard model of particle physics. Fundamentally important is that the term $\boldsymbol{R}^{-1}\partial_\nu\boldsymbol{R}$ cannot give rise to any curvature and as such it cannot be determined by any field equation. Having no propagation, it will not be constrained by either causality or locality restrictions.

Any measurement interrupting the flipping of the spin makes both spinors locked to a definite spin state even if the two spinors have space-like separation, as Goldstone states are made to vanish everywhere simultaneously.

This model of entangled spins for spinor fields may as well be enlarged to incorporate an analogous model of a number of entangled polarizations for photons. In order to see how this is possible, we will consider the formalism presented in [16] and recalled above. Considering a vector field $A_\nu$ that is solution of Maxwell equations in vacuum, we have that it generally consists of a plane wave with an oscillation that could be taken to have place in the plane orthogonal to the second axis, for example picking $A_x$ as the only non-zero

component. The electric and magnetic fields would oscillate in such a plane. A rotation of type (132) would then give rise to the following expression

$$A_a = A_x \begin{pmatrix} 0 \\ \cos \zeta \\ 0 \\ \sin \zeta \end{pmatrix} \tag{138}$$

where we have set $\xi_1^x = 1$ for simplicity. Both electric and magnetic fields now display a rotation in the same plane.

The mechanism is identical to the one above, and it is important to specify that both for the spin in the spinor case and for the potential in the vector case, it is not the coordinate vector (Greek indices) but the Lorentz vector (Latin indices) that is taken as observable quantity. This difference is fundamental, since observable quantities are described by invariants for the passive transformations.

*3.2. Quantum Potentials*

Having described a possible mechanism explaining the feasibility to entangle spins and polarizations in terms of the Goldstone states of spinors and vectors, we now move to a related subject, namely the quantum potentials.

Let us then rewrite the Dirac equations in the polar form (76) and (77) according to

$$Y_\mu - P^\iota u_{[\iota} s_{\mu]} + m s_\mu \cos \beta = 0 \tag{139}$$
$$Z_\mu + P^\rho u^\nu s^\alpha \varepsilon_{\mu\rho\nu\alpha} - m s_\mu \sin \beta = 0 \tag{140}$$

where $2Y_k = (\nabla \beta - 2XW + B)_k$ and $-2Z_k = (\nabla \ln \phi^2 + R)_k$ are what we will call quantum potentials. To see why, let us first combine the Dirac equations in polar form to obtain

$$P^\rho = m u^\rho \cos \beta + \left(Y \cdot u g^{\rho\alpha} - Y^\alpha u^\rho + Z_\mu u_\nu \varepsilon^{\mu\nu\rho\alpha}\right) s_\alpha \tag{141}$$

showing that this is in fact the momentum of the field in its most general expression. In it, we find the kinematic momentum $m u^\rho$ and a number of corrections. One is due to the chiral angle $\cos \beta$ expressing the effects of internal dynamics [15]. The others, proportional to the spin, are given in terms of the $Y_\nu$ and $Z_\mu$ potentials. These contain external contributions of $W_\alpha$ (torsion) and $R_{ij\alpha}$ (gravity) plus the derivatives of the $\beta$ and $\ln \phi^2$ and as such, they are indeed the quantum potentials in relativistic version with spin. The fact that they are at first-order differential is consequence of the relativistic essence and the existence of a second quantum potential is the consequence of internal structure coming from spin.

Because (141) is simply the relation tying the components of the momentum to the derivatives of the degrees of freedom of the spinor, it is merely the Hamilton–Jacobi equation in relativistic form with spin. Thus, it describes the motion of the ensemble of the trajectories, precisely as it is supposed to do within the known de Broglie–Bohm formulation. The difference is that while in the dBB form the module is supposed to guide the particle, which have to be taken as additional entities a posteriori postulated, here we would like to postulate no entity so to allow the interpretation of the particle as the peak of the localized module, and then show that even with no information on the module there can still be guidance. The condition of having a localized module could be easily accommodated in a theory in which the spinor interacts with torsion in its effective approximation [12]. Then, having the module localized means that we can conceal within the material distribution all information about internal dynamics and therefore we may require $\beta$ and the spin to vanish. Hence we have $P^\rho = m u^\rho$ as guidance. With no electrodynamics

$$u_\mu = \frac{q}{m} \partial_\mu \zeta = \frac{1}{m} \partial_\mu S \tag{142}$$

showing that the motion of particles follows trajectories that are guided by the gradient of the phase, or the action functional, recovering the results of the de Broglie–Bohm theory. Consequently the superposition of actions would entail the pattern of interference, precisely as necessary to obtain results of double-slit experiments. Furthermore, for situations in presence of electrodynamics, magnetic fields affect interference as seen in the Aharonov–Bohm effect.

Again, this action functional, that is the phases of the spinorial fields, is what accounts for the information that can be transferred to the gauge and as such it is just the Goldstone state of the unitary transformations.

## 4. Comments

To date, we have seen that, when a spinor field or a vector field are written in their polar form, the various components are rearranged in such a way that the true degrees of freedom remain isolated from the components that can be transferred into the frame or the gauge and which as such are recognized to be the Goldstone states for these fields. Combined to spin connection and gauge potentials these Goldstone states become longitudinal components for the $R_{ji\mu}$ and $P_\mu$ objects. These are proven to be a real tensor and a gauge-invariant vector, so that they possess the same information of both spin connection and gauge potential while being generally covariant as well as gauge-invariant. These tensorial connections are demonstrated to encode information about physical effects although not determined by any field equation with external sources.

The general form of a spinor is therefore given by (49) in terms of module and chiral angle and with the **L** matrix containing Goldstone states. In the subsequent section, we have henceforth used these Goldstone states for the description of a mechanism in terms of which a pair of spinors having opposite spin orientations were connected, or correlated in terms of (131) with $\zeta = \omega t$ and $\omega$ being a generic parameter. So, a mechanism of spin entanglement was presented, where the spin states exhibited a uniform rotation until the moment of measurement but upon the observation the flipping would stop making both spinors locked to either one of the two spin states simultaneously, thus acting as non-predetermined hidden variables. Notice that the lack of predetermination is intended as the non-constancy of the hidden variables. That these variables are hidden is understood by the fact that Goldstone states are expected not to be observable in gauge covariant theories. Furthermore, that these Goldstone states have no determined propagation is the reason of the fact that they do not have to obey locality restrictions. Notice also that this type of non-locality exhibits all properties that Bell non-locality should have, in the sense that: (1) it does not depend on the separation; (2) it is instantaneous; (3) it is monogamous, in the sense that it takes place exclusively between the entangled particles, solutions of the same Dirac equation. The collapse of spinors to either of the definite spin states may occur non-locally with no violation of any relativistic principle. In a more general perspective, the fact that $\zeta = \omega t$ with $\omega$ undetermined and displaying an explicit dependence on the time $t$ means that the collapse is determined by the experimental setting at the very moment of the collapse itself. Therefore, the collapse of spinors to a definite spin state occurs not only non-locally but also contextually to that experimental setting [5,6]. This model was then extended to the case of polarization entanglement, in the case of vector fields. This too had to be expected, since the real and complex Lorentz transformations are defined in terms of the very same parameters and as such the Goldstone states are the same whether a spinor or a vector field is considered. Finally, we recalled how in the usual formulation of the dBB theory one may explain double-slit experiments by means of interference of action functionals, that is, in terms of phases of spinor fields, and so it is the Goldstone states of abelian transformations.

## 5. Conclusions

In this work, we provided a model of spin entanglement in which the Goldstone states of Lorentz transformations are used as non-predetermined non-local hidden variables

compatible with relativity, and we extended it to the case of polarization entanglement. We also recalled that in the dBB theory the explanation of double-slit experiments is achieved by exploiting the Goldstone states of the abelian transformation instead. Either way, there seem to be remarkable applications for Goldstone states in quantum mechanics.

These Goldstone states are additional variables and not observable as expected in any gauge covariant theory. They are not bound by restrictions of locality since their propagation is not determined by any field equation and thus they are fully compatible with relativity. In the present model, we can see how non-locality is compatible with relativity by considering that even if locality pertains to a field that is the solution of relativistic field equations, not all fields have to be solutions of relativistic field equations. In the case of spinors, in fact, the spinor field can be written in terms of the three objects, $L$, $\phi$, and $\beta$, and $L$ remains undetermined in general, even if both $\phi$ and $\beta$ are determined as solutions of field equations. Of these three objects, $\phi$ and $\beta$ constitute the known objective elements of the spinor and $L$ must then be thought as what completes the wave function. In this sense, $L$ might be regarded as what Einstein, Podolsky, and Rosen thought to be the missing element of quantum mechanics, the one whose inclusion would render the description of physical reality complete. Nevertheless, we retain it is philosophically inaccurate to say that $L$ is to be included within the wave function. In fact, $L$ does not have to be included, as it is already there in the most general case, and it would be sufficient not to neglect it.

What the present toy model points out is the potential usefulness of the Goldstone states in assessing problems related to the use of non-local hidden variables in experiments regarding the nature of quantum mechanics.

**Funding:** This research has received no funding.

**Institutional Review Board Statement:** Not applicable.

**Informed Consent Statement:** Not applicable.

**Data Availability Statement:** Not applicable.

**Acknowledgments:** I wish to thank Marie-Hélène Genest for the useful discussions.

**Conflicts of Interest:** The author declares no conflict of interest.

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
