# Peer review of "Goldstone States as Non-Local Hidden Variables"

_universe, doi:10.3390/universe8050277_

Round 1
Reviewer 1 Report
This paper intensively uses the results of previous works [12,13] of the same author. The new result is separation of true spinorial degrees of freedom from the Goldstone states which the author proposes to interpret as the hidden variables. An answer to the question whether this proposition is correct can be done by farther investigations but the idea is interesting and the work certainly should be published in the Universe as it stands now.
Author Response
Dear Referee,
many thanks for your appreciation.
Reviewer 2 Report
Although I have uploaded my review file. I am being asked to submit a review here also. Please see that attached file.

Reviewer 3 Report
In the paper GOLDSTONE STATES AS NON-LOCAL HIDDEN VARIABLES, the spinor and vector fields are considered in polar representation. This allows to reveal features of the Goldstone states as hidden variables, on which the restrictions following from the principles of locality and causality are not imposed. The author analyzes their role in the mechanism that forms the dynamics of nonlocal correlation of entangled states of photons and Dirac particles. The results presented in the paper give a better understanding of the hidden variables problem in relativistic quantum-field systems. I believe that it deserves publication in the journal Universe.
Author Response
Dear Referee,
many thanks for your appreciation